# Imaging the kinetics of anisotropic dissolution of bimetallic core–shell nanocubes using graphene liquid cells

Lei Chen[1,2], Alberto Leonardi [3], Jun Chen[2], Muhan Cao[2], Na Li[4,5], Dong Su [4], Qiao Zhang [1✉], Michael Engel [3✉] & Xingchen Ye [2✉]

Chemical design of multicomponent nanocrystals requires atomic-level understanding of reaction kinetics. Here, we apply single-particle imaging coupled with atomistic simulation to study reaction pathways and rates of Pd@Au and Cu@Au core-shell nanocubes undergoing oxidative dissolution. Quantitative analysis of etching kinetics using in situ transmission electron microscopy (TEM) imaging reveals that the dissolution mechanism changes from predominantly edge-selective to layer-by-layer removal of Au atoms as the reaction progresses. Dissolution of the Au shell slows down when both metals are exposed, which we attribute to galvanic corrosion protection. Morphological transformations are determined by intrinsic anisotropy due to coordination-number-dependent atom removal rates and extrinsic anisotropy induced by the graphene window. Our work demonstrates that bimetallic core-shell nanocrystals are excellent probes for the local physicochemical conditions inside TEM liquid cells. Furthermore, single-particle TEM imaging and atomistic simulation of reaction trajectories can inform future design strategies for compositionally and architecturally sophisticated nanocrystals.

[1] Institute of Functional Nano & Soft Materials (FUNSOM), Jiangsu Key Laboratory for Carbon-Based Functional Materials and Devices, Soochow University, 215123 Suzhou, China. [2] Department of Chemistry, Indiana University, Bloomington, IN 47405, USA. [3] Institute for Multiscale Simulation, IZNF, Friedrich-Alexander University Erlangen-Nürnberg, 91058 Erlangen, Germany. [4] Center for Functional Nanomaterials, Brookhaven National Laboratory, Upton, NY 11973, USA. [5] Frontier Institute of Chemistry, Frontier Institute of Science and Technology jointly with College of Science, Xi'an Jiaotong University, 710054 Xi'an, Shanxi, China. ✉email: qiaozhang@suda.edu.cn; michael.engel@fau.de; xingye@indiana.edu

D irect imaging of nanoscale reaction kinetics in the liquid phase is vital for developing a comprehensive mechanistic understanding of these processes[1,2]. Over the past decade, in situ liquid cell electron microscopy has emerged as a powerful tool for unlocking previously unseen and inaccessible kinetics underlying nanocrystal growth[3–17], dissolution[18–23], transformation[24–28], and self-assembly[29–36] on the relevant length and time scales. Controlled etching or dealloying of multimetallic nanocrystals has been used as a strategy towards nanostructures with shape and composition otherwise impossible to access by bottom-up synthetic methods[37–43]. Our current understanding of how these nanoscale sculpting processes occur has largely relied on methods that separate intermediates from the reaction medium and characterize them ex situ[44,45]. However, short-lived intermediates are often difficult to probe with this approach. Recently, several studies imaged dissolution events on individual monometallic nanocrystals with in situ liquid cell TEM techniques[18,19,21]. Using premade nanocrystals with well-defined shapes allows the three-dimensional morphology of reaction intermediates to be inferred from their two-dimensional projected images and, in some cases, time-lapse particle volume and surface facets to be deduced[19,21]. Nonetheless, reaction pathways and dissolution kinetics of multimetallic nanocrystal, whether being a core–shell structure or an alloy, remain largely unexplored.

While in situ TEM observations can provide important insight into processes in a liquid environment, the high-energy electron beam inevitably perturbs many processes under investigation. The realization that radiolysis generates a number of transient radical species that can activate or suppress redox chemistries in solution has motivated systematic studies to determine the steady-state distribution of beam-induced species under different TEM-imaging conditions[23,46,47]. So far, the impact of a physically confined space created by encapsulating windows, such as graphene and silicon nitride membranes on the physicochemical processes under in situ TEM conditions remain much less understood.

Herein, we report graphene liquid cell (GLC) TEM imaging of oxidative dissolution of core–shell nanocubes composed of dissimilar metals. Our motivation is twofold. First, the three-dimensional shape and the elemental contrast of core–shell nanocubes visualized along the etching trajectory yields valuable information about the complex environment within GLCs when compared with monometallic nanocrystals. Second, a detailed mechanistic understanding of how the distinct reactivity between core and shell metals influences shape transformation pathways and dissolution kinetics is at present not available due to the lack of systematic single-particle experimental studies.

## Results

**In situ TEM imaging of Pd@Au nanocube dissolution.** We chose nanocubes as the model system in this study because their {100} surface facets ensure the alignment of the vast majority of nanocubes along their <100> zone axis. This preferred particle orientation is critical for deducing crystallographically meaningful information from complex shape intermediates during dissolution of core–shell nanocubes. Bimetallic Pd@Au nanocubes were synthesized using a seed-mediated method by depositing epitaxial Au shells onto preformed Pd nanocubes. The difference in atomic number (Pd: 46 and Au: 79) allowed the core and the shell domains to be distinguished and their dissolution kinetics to be quantified with in situ bright-field TEM imaging. Scanning-TEM (STEM) characterization revealed that the cuboidal shape of Pd seeds was well retained after Au shell growth (Fig. 1a, b,

Supplementary Fig. 1). Intermixing between Pd and Au, if any, was limited to the core–shell interface (Fig. 1c–e).

We used GLC TEM to monitor reactions taking place on individual nanocubes (Fig. 2a)[6]. An aqueous mixture of nanocrystals and ferric chloride ($FeCl_3$) was encapsulated between two sheets of graphene supported by holey carbon film on gold TEM grids. The concentration of $FeCl_3$ was optimized such that controlled dissolution of nanocrystals can be initiated upon localized electron beam illumination. Particle etching was

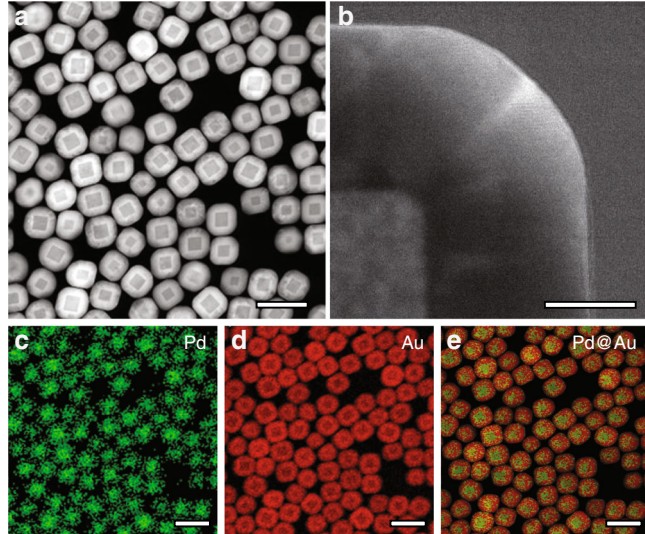

**Fig. 1 TEM characterization of Pd@Au core–shell nanocubes. a** Low-magnification and **b** high-magnification STEM-HAADF images of Pd@Au nanocubes (44-nm core and 35-nm shell). **c–e** STEM-EDX results of Pd@Au nanocubes with elemental maps for Pd **c**, Au **d** and their overlay **e**. Scale bars: 200 nm **a**, **c–e** and 20 nm **b**.

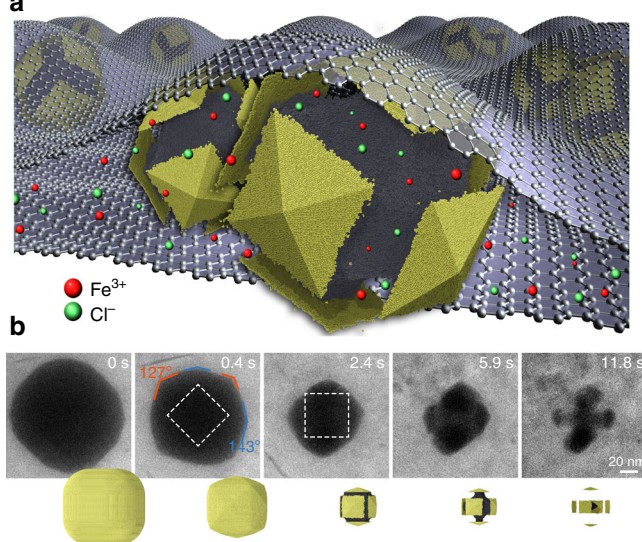

**Fig. 2 Real-time imaging of rapid dissolution of a Pd@Au nanocube inside a GLC. a** Schematic illustration of a GLC encapsulating a solution of nanocubes and oxidative etchants. Carbon atoms of graphene sheets are enlarged for clarity purpose. **b** Time-lapse TEM images and corresponding simulation snapshots extracted from Supplementary Movies 1a and 4 showing the dissolution of nanocubes (44-nm Pd core and 35-nm Au shell) under strongly oxidative environment. Chemical potential in simulation: −3.600 eV.

negligible before exposure to the electron beam. The primary oxidative species responsible for sculpting of nanocubes is believed to be the hydroxyl radical (•OH), which is a radiolysis product of aqueous solutions when irradiated with high-voltage electron beam[19,21]. The hydroxyl radical is a potent oxidizing agent with an electrochemical potential of 2.80 V versus the standard hydrogen electrode[10,23,46]. Experimentally we found that the rate of nanocrystal etching was mainly determined by the electron beam dose rate. This observation is consistent with previous studies on monometallic Au nanocrystals where the electron flux controls the total concentration of oxidative species in solution while the concentration of ferric ions likely modulates the chemical potential by changing the relative abundance of various beam-generated oxidative species[19,23].

Figure 2b presents time-lapse images for a Pd@Au nanocube (44-nm core and 35-nm shell) undergoing rapid oxidative etching (Supplementary Movie 1a). The cubic Au shell transformed into square pyramids bounded by vicinal {hk0} facets, as evidenced by the well-defined octagonal cross section with alternate interior angles captured for this intermediate shape[48]. The apex angles of individual pyramids were measured from selected frames of the video to deduce {hk0} facet indices. At $t = 0.4$ s, the Au shell was determined to be terminated predominantly by {210}-type facets. The {210} vicinal planes consist of two-atom wide {100} terraces separated by monoatomic steps. A qualitatively similar transformation pathway was observed when imaging the dissolution of Au-only nanocubes, although a tetrahexahedron (THH) enclosed by {310} vicinal facets was found to be the steady-state shape intermediate (Supplementary Fig. 2). The high volumetric etching rate of Au (ca. 20 million atoms/s) is indicative of non-equilibrium conditions under which formation of {hk0} facets from initially flat {100} crystal planes can be attributed to the step-recession mechanism[19]. In-plane rotation of intermediate nanocrystal was observed at 3.0 s. Exposure of the Pd core region to the etching solution began at ~2.4 s, after which the Au shell fragmented into disconnected pyramidal domains. Dissolution of the Pd core was faster than that of the Au shell once both metals came in contact with the etching solution, producing a complex architecture in which several Au branches were connected through a central Pd core. A hexapodal structure was readily resolved by tilting the TEM specimen holder prior to initiation of the etching reaction (Supplementary Movie 1b). This intermediate shape derives from the intrinsic anisotropy in the dissolution of Au shell exposing selectively 12 edges of the Pd cube to etchants[49]. The transformation pathway including the sequence of intermediate shapes is corroborated by Monte Carlo (MC) simulations of etching Pd@Au nanocubes (Fig. 2b, bottom row).

**Dissolution kinetics**. To better visualize the intermediate shapes and to obtain a quantitative understanding of the dissolution kinetics, we carried out slow oxidative dissolution on Pd@Au nanocubes by controlling the electron beam dose rate such that the total reaction duration was prolonged to 190 s. Time-lapse TEM images from Supplementary Movie 2a and corresponding simulation snapshot are presented in Fig. 3a. The nanocube did not visibly rotate during the first 100 s of reaction and therefore the <100> zone axis was maintained, and a bimetallic multipod was observed after 137 s. We constructed time-domain contour plots by extracting nanocrystal outlines from individual movie frames and color-coding them based on calculated local curvature (Supplementary Fig. 3b, Supplementary Movie 2b). This spatiotemporal reactivity map indicates that etching was faster at those high-curvature and presumably high-energy sites.

The consequence is the appearance of {210} vicinal planes as steady-state surface termination for the Au shell. As illustrated in Fig. 3d, etching of peripheral edge atoms on a series of receding {100} terraces was clearly resolved on a dissolving Pd@Au nanocube. During this period, etching along the orthogonal direction was much slower. The step-recession process was routinely observed on multiple side-facets of Pd@Au nanocubes (Supplementary Movie 3), demonstrating that the formation of THH-like shape intermediate is a generalized and highly reproducible outcome when metallic nanocubes are subject to oxidative dissolution under non-equilibrium conditions. The step-recession can be rationalized by considering coordination number (CN)-dependent reactivity where atoms at step edges with CN = 6 are deleted more rapidly than atoms at terrace sites with CN ≥ 7. The type of {hk0} facet is a kinetic equilibrium between lateral step recession and dissolution of surface {100} terraces along the orthogonal direction (Supplementary Fig. 3a). MC simulations show that such steady-state non-equilibrium shape is directly influenced by the reaction environment (Supplementary Fig. 4), and increasing strength of the oxidizing environment leads to faster dissolution of {100} terraces and more gradual pyramidal features (Supplementary Fig. 4b).

We analyzed the dissolution kinetics of Pd@Au nanocubes within the framework of the Lifshitz, Slyozov, and Wagner (LSW) theory originally developed to describe crystal growth[50,51]. According to LSW theory, the etching behavior of crystalline materials can be classified into three regimes[20]. In the diffusion-limited regime, a linear relationship between particle volume $V$ and reaction time $t$ is expected. This situation arises under corrosive conditions where the generation of oxidative species is rate-limiting and surface atoms are removed at a constant rate regardless of their reactivity. Conversely, if etching is limited by surface reactions and the same number of atomic layers is removed within each time interval regardless of particle size, a linear dependence of $V^{1/3}$ vs. $t$ would result. The intermediate scenario corresponds to a constant removal rate of rows of atoms yielding a linear dependence of $V^{2/3}$ vs. $t$. Such etching kinetics is favored when atoms at the step edge are removed preferentially over those at terrace sites.

The volume of Au shell was estimated frame-by-frame to analyze dissolution kinetics (Supplementary Fig. 5). Linear regression analysis revealed that the plot of $V^{2/3}$ vs. $t$ gave a more satisfactory fit to the data points prior to exposure of the Pd core (up to about 100 s), whereas the plot of $V^{1/3}$ vs. $t$ yields a better fit to the data when Au and Pd were simultaneously exposed to the etching solution (Fig. 3b). These experimental kinetics data are quantitatively reproduced by MC simulations (Fig. 3c), indicating that the mechanism of etching changes from edge-selective toward layer-by-layer removal of atoms. It is worth noting that the volume of Au shell after its breaking into islands was estimated based on domains that derive from side-facets of the Pd@Au nanocube.

**Intrinsic and extrinsic factors responsible for anisotropic dissolution**. Shortly after the Pd core became exposed to etchants at about 105 s, an out-of-plane rotation of the particle occurred (Fig. 3a). Two important structural features were recognized from the TEM image recorded at 137 s immediately after particle rotation. First, the upper and lower (orientation shown on the image) Au pyramids were formed via dissolution of the lateral nanocube facets. Second, the left and right (orientation on the image) Au domains, which resulted from etching of the initial top and bottom nanocube surfaces, have larger volumes than other Au domains. In the rest of this paper, "top and bottom" facets refers to the two nanocube facets that are orthogonal to the

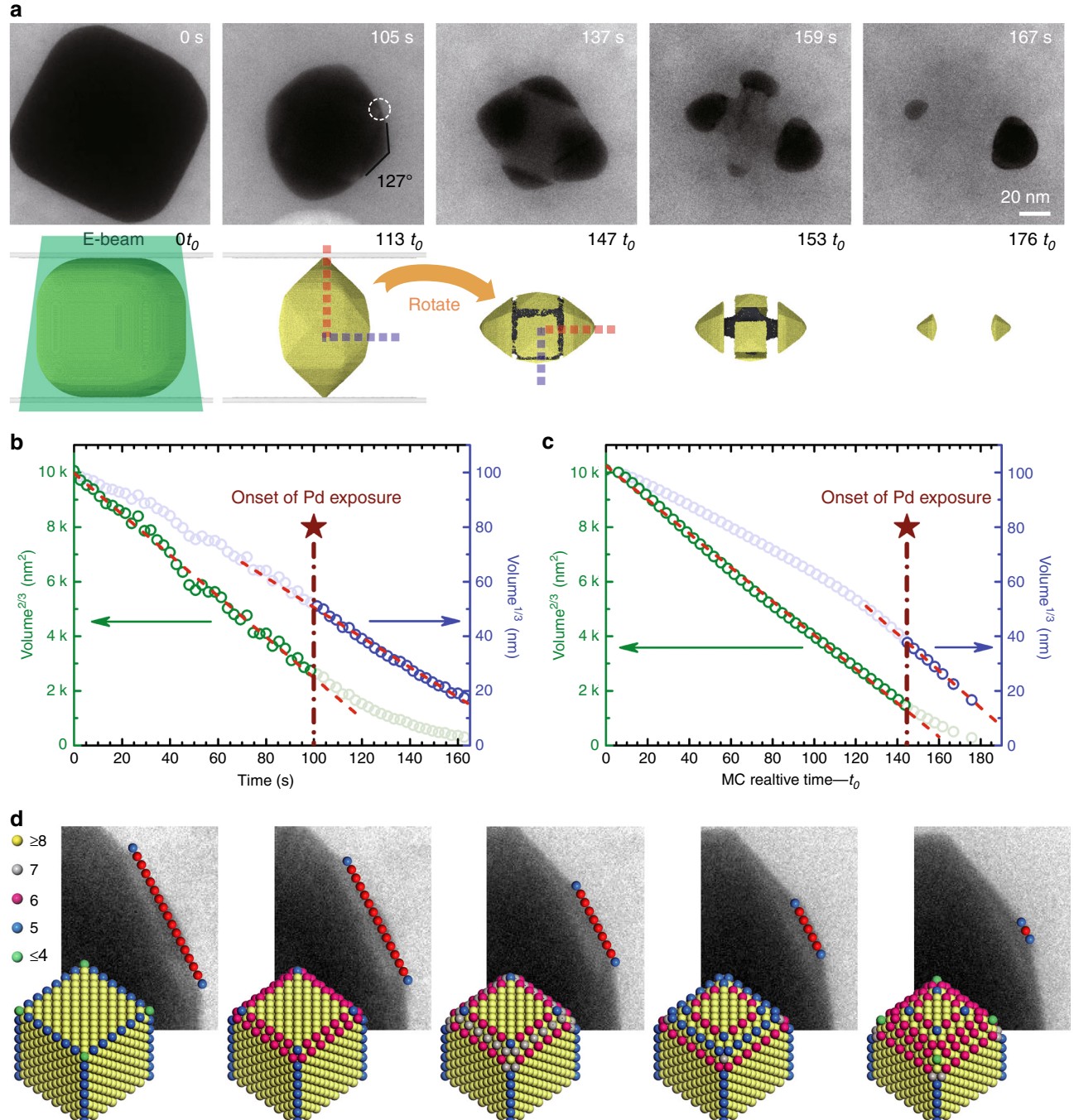

**Fig. 3 Dissolution kinetics and reaction pathway of a Pd@Au nanocube inside a GLC. a** Time-lapse TEM images and corresponding simulation snapshots extracted from Supplementary Movies 2a and 4 showing slow dissolution of nanocubes (44-nm Pd core and 35-nm Au shell) when exposed to a weakly oxidative environment. Chemical potential in simulation: −3.525 eV. **b, c** Au-shell volume to the power of 1/3 and 2/3 as a function of time based on data from experiment **b** and simulation **c**. **d** Top: Time-lapse TEM images extracted from Supplementary Movie 3 showing the step recession process responsible for the evolution of pyramidal feature from an initially flat (100) facet of the Au shell. Bottom: Atomistic models for a Pd@Au nanocube undergoing oxidative dissolution. Individual atoms are color-coded according to coordination number with color-coded legends shown at top left.

electron beam before etching starts. This dissolution behavior was unexpected and would have been difficult to visualize if monometallic Au cubes were used.

While slower etching of the top and bottom facets compared to lateral facets is commonplace when imaging the dissolution of Pd@Au nanocubes, MC simulations in a homogenous solution not accounting for extrinsic factors failed to reproduce the difference in etching rates (Supplementary Movie 4). Indeed, such kinetic anisotropy cannot be explained by intrinsic anisotropy in

reactivity for highly symmetric objects, such as nanocubes made of face-centered cubic metals. It should result, instead, from anisotropy induced by the GLC setup. A similar dissolution asymmetry was not observed when we attempted etching of monometallic Au nanocubes using GLC (Supplementary Movie 5). One plausible explanation is that the Au-only nanocubes used here or studied previously were much smaller (45–60 nm edge length) than the core–shell Pd@Au nanocubes (110–130 nm edge length).

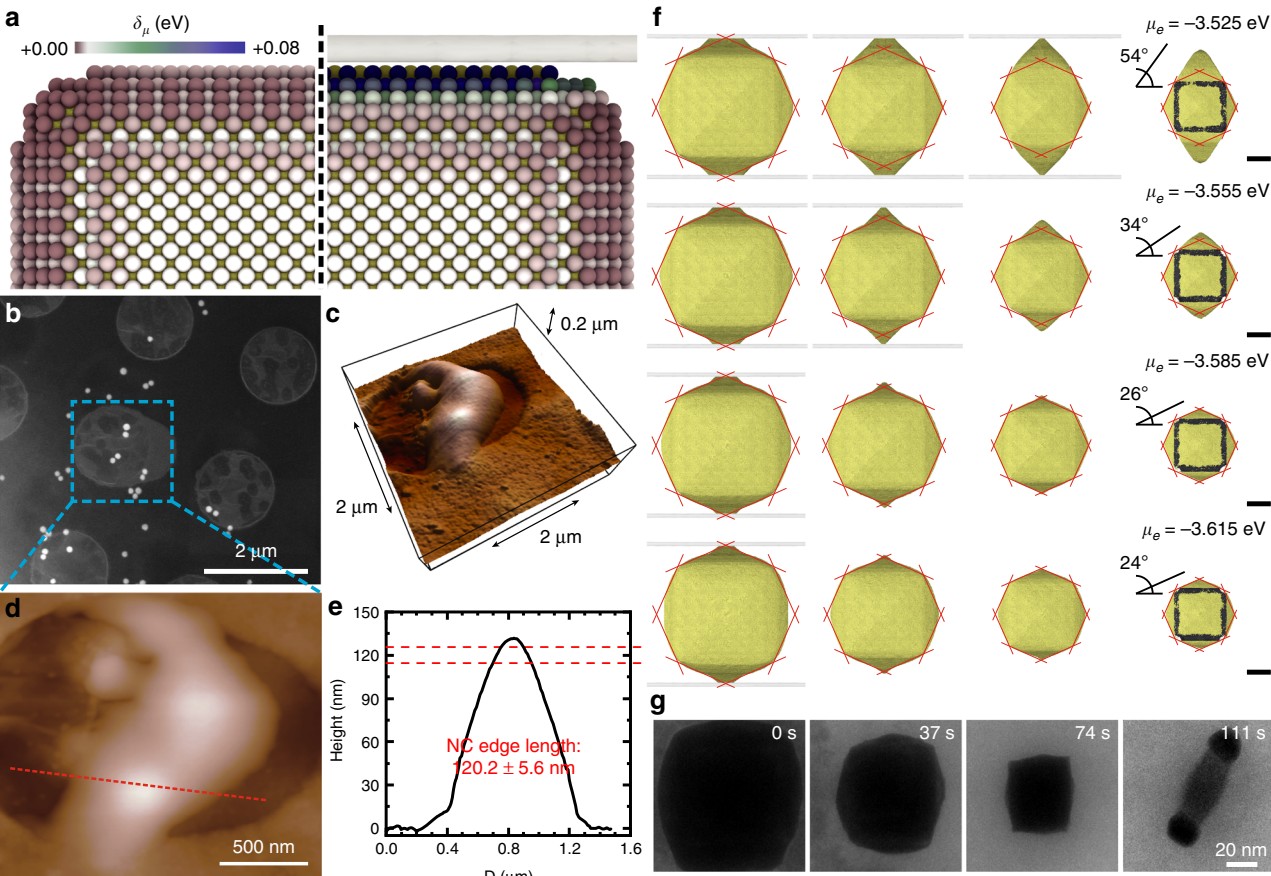

**Fig. 4 Dissolution anisotropy of Pd@Au nanocubes induced by graphene encapsulation. a** Effective chemical potential for surface Au atoms of a Pd@Au nanocube subject to isotropic (left) and anisotropic (right) oxidative environment due to a graphene sheet. **b**, **c** Correlative SEM **b** and AFM **c** images of the same GLC pocket. **d** High-resolution AFM height image of the region indicated in **b**. **e** Height profile through the red dashed line indicated in **d**. **f** Simulations of anisotropic dissolution of Pd@Au nanocubes induced by top and bottom graphene sheets at different chemical potentials. **g** Time-lapse TEM images extracted from Supplementary Movie 6 showing the formation of a rod-like nanostructure due to preferential lateral dissolution of Pd@Au nanocubes (70-nm Pd core and 22-nm Au shell).

The multi-layer graphene membranes supposedly wrap around the flat top and bottom {100} facets of the nanocubes as they seal to form liquid pockets. To test this hypothesis, we carried out correlative electron microscopy and atomic force microscopy (AFM) studies to determine the thickness of individual GLCs with trapped Pd@Au nanocubes. A representative dataset is shown in Fig. 4b–e and Supplementary Fig. 6. From AFM height analysis, the thickness of this GLC was determined to be 130 nm, which is similar to the edge length of as-made Pd@Au nanocubes ($120.2 \pm 5.6$ nm). The potentially tight interface or close proximity between nanocube and the graphene sheets can limit the accessibility of diffusing oxidative species. Consequently, the effective local chemical potential experienced by the top and bottom nanocube facets is expected to be different from that experienced by its side facets (Fig. 4a). In a recent study, the gapless interface between graphene sheets and colloidal nanocrystals, although based on GLCs that were nearly dried-out, was confirmed via direct cross-sectional imaging of the heterointerface by using high-resolution aberration-corrected TEM[52]. Nevertheless, a range of liquid layer thicknesses among various GLC pockets likely existed in our sample. To reduce the degree of anisotropy caused by extrinsic factors, such as the finite thickness of GLCs, ongoing work in our laboratory includes the use of spacer particles during GLC preparation to create a more isotropic environment for the nanocrystals under study.

To quantitatively describe the blocking effects of confining graphene sheets, we performed MC simulations with less-reactive top and bottom sides of the nanocube to mimic the anisotropic reaction environment. In these simulations, we introduced an atomic bond energy that varies with CN to represent local surface conditions. A linear relationship was consistently found between (volume of Au shell)$^{2/3}$ $V^{2/3}$ and time $t$ prior to exposure of the Pd core at different chemical potentials (Supplementary Fig. 7). These kinetics results from simulation suggest that the graphene windows do not alter the edge-selective mechanism of etching. Another experimental observation is that the {100} facets regress while the vicinal {hk0} facets develop, which is also closely matched by our simulations. For simulations performed in isotropic environment (i.e., without graphene windows), the surface termination delimiting the THH shape shows Au pyramids with a steady-state protrusion angle (i.e., the angle between the face of a pyramid and the square base) of 24° (Supplementary Fig. 8). This angle corresponds to THH facet indices intermediate between {210} and {310} and is in line with previous studies[19]. Figure 4f shows simulation results for etching of Pd@Au nanocubes taking into account the graphene blocking effects. It was found that the strength of oxidizing environment determines whether the Pd core is exposed before the formation of THH-like pyramidal features. With a more positive chemical potential (i.e., less oxidative), the surface facets of THH-like Au domains deviated dramatically from simulations performed in

isotropic environment. The oxidative strength essentially modulated the steady-state protrusion angle of the square pyramids, which can vary from 24° up to 66° (Fig. 4f). The slower etching of top and bottom Au shells protected the core from etchants along these directions, which sometimes even resulted in the formation of a Au–Pd–Au segmental nanorod (Fig. 4g and Supplementary Movie 6). The formation of the rod-shaped intermediate confirmed once more that the etching rate on side directions was uniform and higher than the etching rate close to the graphene sheets.

**Effects of dissimilar reactivity between core and shell metals on dissolution kinetics**. Another recurring observation was that etching of the Au shell slowed down once the Pd core became exposed to the oxidative environment. Notably, in the case of fast etching of Pd@Au nanocubes, 90% of Au shell was dissolved within 2.4 s of reaction time. It took another 10 s to dissolve the residual 10% volume (Fig. 2b). The slowdown in dissolution rate suggests that Pd and Au do not respond independently to the oxidative environment. Instead, electronic coupling between the two domains plays an important role in dictating the dissolution behavior of bimetallic nanocrystals. The slowdown in Au etching is also manifested in the kinetics of slow oxidative dissolution and resulting particle shape evolution. From the $V^{1/3}$ vs. $t$ plots in

Fig. 3b, c, we observed that dissolution of Au slowed down slightly compared to predictions from MC simulations upon exposure of the Pd core. Moreover, the well-faceted Au pyramids present at $t = 137$ s had transformed into more rounded shapes by $t = 159$ s (Fig. 3a). Selective dissolution of Pd along the Pd/Au interface and fast etching of peripheral Au atoms lead to edge rounding at the basal plane of the THH Au islands.

To further assess the influences of Pd core on the etching kinetics of Au shell, we performed GLC imaging of rapid dissolution of thin-shell Pd@Au nanocubes with nearly identical overall volume (Fig. 5a, Supplementary Movie 7a). This 70-nm core and 22-nm shell Pd@Au nanocube, which took merely 10 s to be completely dissolved, also yielded isolated THH-like Au pyramids as the steady-state non-equilibrium intermediate. The top and bottom islands were etched much more slowly than the lateral ones (e.g., Fig. 5a, frame at $t = 5.1$ s). Both $V^{2/3}$ vs. $t$ and $V^{1/3}$ vs. $t$ plots for the Au shell exhibited an abrupt slope change at around 0.6 s, which is well-correlated with the beginning of etching of the Pd core (Fig. 5b). A similar slowdown in Au etching was observed while imaging the dissolution of Pd@Au nanocubes with different Au-to-Pd volume ratios (Supplementary Figs. 9, 10 and Supplementary Movies 7b, 8 and 9). It was a general observation that Au islands remained even after Pd was fully dissolved. The sudden deceleration of Au etching was not

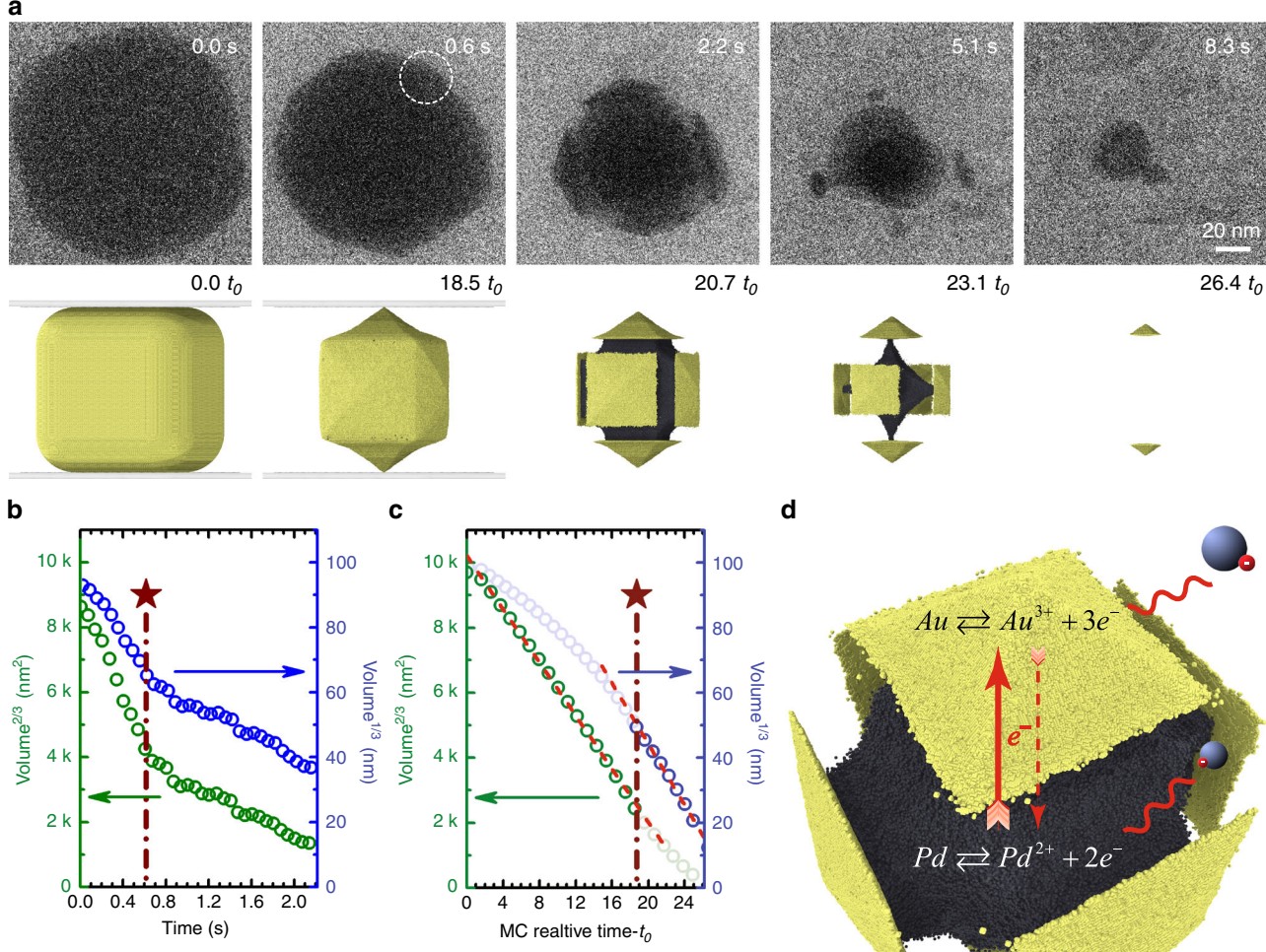

**Fig. 5 Rapid dissolution of a thin-shell Pd@Au nanocube. a** Time-lapse TEM images and corresponding simulation snapshots extracted from Supplementary Movies 4 and 7a showing rapid dissolution of thin-shell nanocubes (70-nm Pd core and 22-nm Au shell) when exposed to a weakly oxidative environment. Chemical potential in simulation: −3.525 eV. **b**, **c** Au-shell volume to the power of 1/3 and 2/3 as a function of time based on data from experiment **b** and simulation **c**. **d** Schematic description of the galvanic corrosion protection mechanism responsible for the passivation of Au domain. The electrons released from oxidation of Pd either react with oxidative species in the liquid environment or migrate to Au.

seen in MC simulations where the probability of atom removal for Pd and Au was set to be determined by the standard electrochemical potential of bulk metals and the CN of individual atoms (Fig. 5c). In other words, the two metals were assumed to respond independently to the oxidative environment in these simulations. We believe the different behaviors between simulation and experiment are caused by contact between the two metals. Considerations of electronic/electrochemical interactions between core and shell metals in simulation are not possible with the current model and will be addressed in future work.

Dissimilar metals that are electrically connected in the presence of an electrolyte form an electrochemical cell. Galvanic corrosion (also known as bimetallic corrosion) occurs where the more active metal (i.e. the anode) corrodes and the less active metal (i.e. the cathode) is protected. Electrons released from oxidation reactions can either recombine with radicals in the solution or migrate to the other metal (Fig. 5d). A net migration of electrons from Pd to the more noble Au develops once a steady state is established. Consequently, the potential in the Au domain is driven in the negative direction and the rate of dissolution slows down. Importantly, the more pronounced slowdown of Au etching observed during rapid dissolution of Pd@Au nanocrystals indicate the presence of a larger migration of electrons from Pd to the Au domain in the steady state.

**Nanobox formation.** To probe the dissolution behavior of bimetallic core–shell nanocrystals with strongly dissimilar reactivity, we synthesized Cu@Au nanocubes with a highly reactive Cu core coated with a 3-nm thin Au shell (Fig. 6a, Supplementary

Fig. 11). As depicted in Fig. 6b, no appreciable shape change occurred during the first few seconds of electron-beam illumination. After that, the Cu core dissolved rapidly (within 0.2 s) leaving behind a Au nanobox with thickness of several atomic layers and hollow interior (image at $t = 3.8$ s). Apparently, localized corrosion at high-energy sites, such as corner defects during the initial quiescent period was followed by eventual permeation of the etchant through the Au shell. Continued etching of the residual Au nanobox proceeded through expansion of primary pinholes at cube corners into larger voids ($t = 4.8$ s). Two levels of contrast were discerned between the time period of 4.2–5.6 s and after 5.6 s, suggesting that opposing facets of the Au nanobox dissolved sequentially forming a nanocage with enhanced porosity (Fig. 6d, Supplementary Movie 10a). Similar dissolution trajectories were observed when a pair of Cu@Au nanocubes were etched under the same conditions (Supplementary Fig. 12a, Supplementary Movie 10b).

MC simulations taking into account the difference in electrochemical potential between Cu and Au qualitatively reproduced the morphological changes observed in experiment (Fig. 6c, Supplementary Movie 11). Importantly, the rapid dissolution of Cu core demonstrates that the etching kinetics is not limited by the generation and diffusion of oxidative species such as (•OH). Instead, dissolution processes in this study are largely determined by composition and reactivity of metal nanocrystals. Finally, we allude to the potential application of GLC TEM to measure shell thickness and uniformity within an ensemble of core–shell nanoparticles by imaging the dissolution behavior of multiple objects in the same field of view

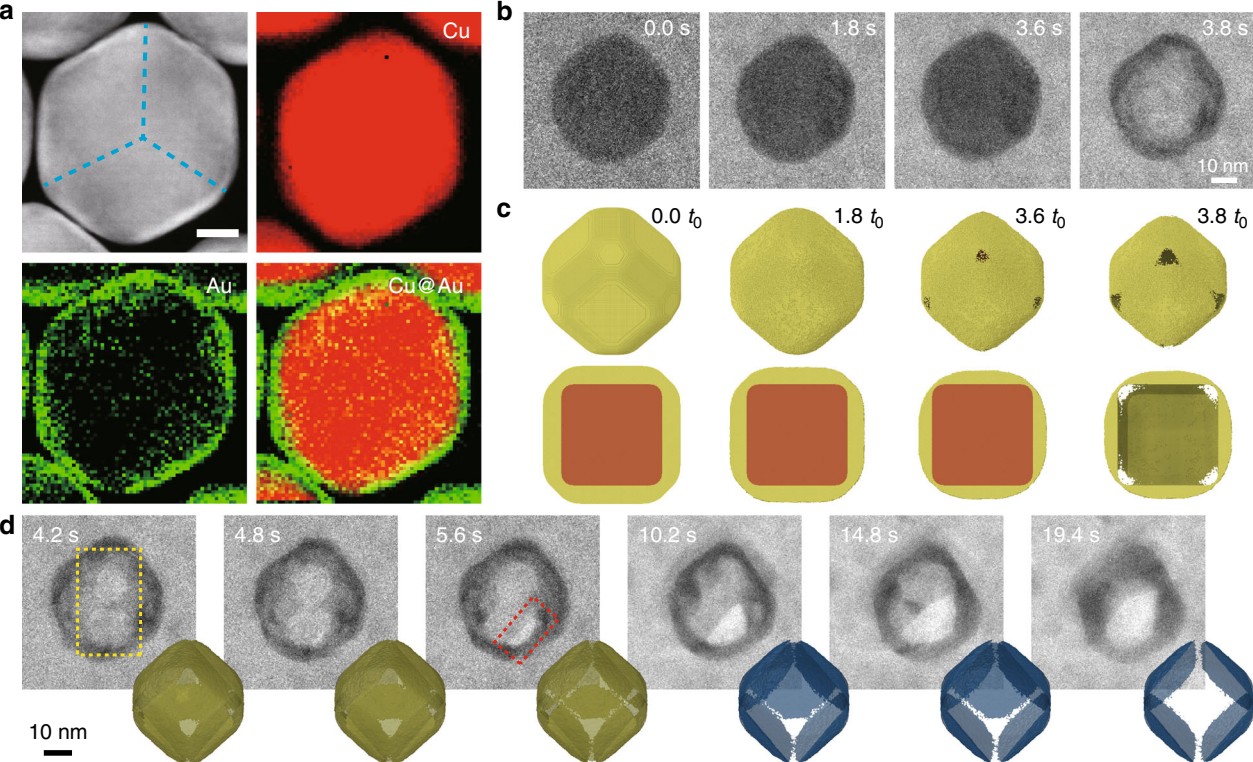

**Fig. 6 Nanobox intermediate during dissolution of truncated Cu@Au nanocube. a** STEM-HAADF image and STEM-EELS elemental maps of a Cu@Au nanocube (44.3-nm Cu core and 2.6-nm Au shell). Scale bar: 10 nm. **b, c** Time-lapse TEM images **b** and corresponding simulation snapshots **c** extracted from Supplementary Movies 10a and 11 showing rapid dissolution of the Cu core of a truncated Cu@Au nanocube yielding a thin-shell nanobox. Cut-section snapshots in the bottom row reveals rapid dissolution of the Cu core upon exposure to the oxidative environment while dissolution of Au shell is negligible. Chemical potential in simulation: −3.525 eV. **d** Time-lapse TEM images extracted from Supplementary Movie 10a showing etching of the Au shell upon continued electron beam irradiation and corresponding Monte Carlo simulation snapshot drawn by rendering nanocrystal shape with transparency. The last three snapshots (blue color) are not derived from simulation but drawn to illustrate the three-dimensional shape.

(Supplementary Fig. 12b), which may benefit the development of complex nanoparticles for fuel cell catalysis, drug delivery, etc.

## Discussion

We investigated the reaction kinetics during dissolution of well-defined bimetallic core–shell nanocubes using real-time GLC TEM imaging and atomistic MC simulations. We uncovered the anisotropy of reaction environment within GLCs due to graphene windows. Kinetic analysis after appropriate accounting for extrinsic anisotropy reveals that etching changes from predominantly edge-selective to layer-by-layer removal of Au atoms during dissolution of Pd@Au core–shell nanocubes. Depending on the Pd-to-Au volume ratio, either an absence of acceleration or a pronounced slowdown in Au shell dissolution was observed once etching reached the Pd core. These results can be rationalized by the galvanic corrosion protection mechanism. Our observations revealed diverse dissolution reaction pathways that provide access to bimetallic multipods, segmental nanorods, nanoboxes, and nanocages via controlled etching of core–shell nanocubes. Understanding and deliberately controlling the physicochemical environment inside liquid cells are critical to transferring the insights gained from in situ TEM study to ex situ colloidal synthesis of nanocrystals.

## Methods

**Chemicals**. Potassium iodide (KI, ≥99.0%), L-ascorbic acid (AA, ≥99.5%), sodium hydroxide (NaOH, ≥98%), hydrogen tetrachloroaurate trihydrate (HAuCl₄·3H₂O, ≥99.9% trace metals basis), silver nitrate (AgNO₃, ≥99.0%), sodium borohydride (NaBH₄, 99%), trioctylphosphine oxide (TOPO, 99%), copper(I) bromide (CuBr, 99.999%), oleylamine (technical grade, 70%), trioctylphosphine (TOP, 97%), iron (III) chloride (97%), hydrochloric acid (HCl, 37 wt% in water), nitric acid (HNO₃, 70%) and sodium persulfate (Na₂S₂O₈, ≥98%) were purchased from Sigma Aldrich. Palladium(II) chloride (PdCl₂, 59% Pd) was purchased from Acros Organics. Hexadecyltrimethylammonium bromide (CTAB, >98.0%) was purchased from TCI America. All chemicals were used as received without further purification. Ultrapure water (18.2 MΩ cm at 25 °C) obtained from a Barnstead® GenPure® water purification system (Thermo Scientific) was used in all experiments. All glassware was cleaned with aqua regia (a mixture of HCl and HNO₃ in 3:1 volume ratio), rinsed thoroughly with water and dried before use.

**Synthesis of Pd@Au core–shell nanocubes**. Pd nanocubes were synthesized according to previously reported method[53]. Briefly, a certain amount (2.5 mL for nanocubes of 22 nm in edge length, and 1.25 mL for nanocubes of 46 nm in edge length) of 20 mM H₂PdCl₄ solution (prepared by dissolving 0.071 g of PdCl₂ in 20 mL of 40 mM HCl solution), 2.5 mL of 0.04 M KI, 1 mL of 0.1 M AA and 1.5 mL of H₂O were added into 12.5 mL of 0.1 M CTAB solution under stirring. Afterwards, the mixture was heated to 90 °C and kept at this temperature for 60 min. After the solution was cooled down, Pd nanocubes were isolated by centrifugation at 8500 rpm for 10 min, and were then dispersed in 5 mL water. To synthesize Pd nanocubes of 70 nm in edge length, 1 mL of 10 mM H₂PdCl₄, 100 μL of 0.1 M NaOH, 10 mL of 2.5 mM CTAB, and 10 μL of 22 nm Pd nanocube solution were added into 0.2 mL of 100 mM AA. The resulting mixture was maintained at 40 °C for 12 h. The products were separated via centrifugation at 4000 rpm for 5 min and were subsequently dispersed in 5 mL of H₂O.

The Pd@Au nanocubes were synthesized by using Pd nanocubes as seed particles. Specifically, for small Pd@Au nanocubes (22-nm Pd core and 5-nm Au shell), 10 mL of 50 mM CTAB, 20 μL of Pd nanocube solution, and 0.5 mL of 0.1 M AA were mixed together under stirring. 50 μL of 10 mM HAuCl₄ was added dropwise to the mixture to start overgrowth of Au shell. To synthesize large core (70 nm) and thin shell (21 nm) Pd@Au nanocubes, 10 mL of 50 mM CTAB, 200 μL of Pd nanocube solution, and 500 μL of 0.1 M AA were mixed together under stirring, to which 400 μL of 10 mM HAuCl₄ was added. For small core (46 nm) and thick shell (38 nm) Pd@Au nanocubes, 80 μL of 0.1 M AA, 200 μL Pd nanocube solution and 10 mL of 50 mM CTAB were mixed under stirring, followed by dropwise addition of 2.5 mL of 10 mM HAuCl₄. All overgrowth reactions were carried out at 25 °C under stirring for 10 min. The nanocubes were purified by centrifugation at 5000 rpm for 5 min and were dispersed in 5 mL of H₂O for in situ TEM studies.

**Synthesis of Cu@Au nanocubes**. Cu nanocubes were synthesized according to previously reported method[54]. Briefly, 1.5 mmol of CuBr, 7.5 mmol of TOPO, and 35 mL of oleylamine were loaded into a 100 mL three-neck flask. After degassing under vacuum for one hour at 100 °C, the reaction mixture was heated rapidly to 220 °C under nitrogen atmosphere and was kept at this temperature for 60 min

before being lowered to 140 °C. Afterwards, a solution consisting of 40 mg of HAuCl₄ dissolved in 3 mL of TOP was added to the reaction flask at a flow rate of 0.5 mL/min by using a syringe pump. The reaction solution was cooled down to room temperature after one hour and the Cu@Au nanocubes were purified via three rounds of precipitation with isopropanol and centrifugation at 6000 rpm for 5 min. The nanocubes were finally dispersed in 5 mL of hexane for in situ TEM studies.

**Preparation of graphene-coated TEM grids**. Three to five layers of graphene on Cu foil was purchased from ACS Material, LLC. 300-mesh Quantifoil gold TEM grids (Structural Probe, Inc. Catalog number: 4230G-XA) were placed on top of a piece of graphene-coated Cu foil with the amorphous carbon side facing graphene. Isopropanol was then dropped onto the Cu foil to bind graphene to Quantifoil TEM grids. Afterwards, the Cu foil with adhered TEM grids on top was floated on a 0.1 g/mL sodium persulfate solution to etch the underlying Cu foil. These graphene-coated TEM grids were rinsed with ultrapure water three times and dried in air before further use.

**Preparation of GLCs**. In a typical process, 80 μL of acidified etching solution composed of 0.15 M FeCl₃ and 0.3 M HCl was mixed with 20 μL of Pd@Au nanocube solution. To encapsulate liquid samples with GLCs, one graphene-coated TEM grid was first laid onto a glass slide with the graphene side facing upwards. Next, a small droplet of the above-mentioned nanocube-etchant solution was placed onto the central region of the TEM grid. Another graphene-coated TEM grid was then placed on top of the droplet, forming sealed GLC pockets within 10 min. TEM imaging was performed within 30 min of GLC formation. For experiments on the etching of Cu@Au nanocubes, a hexane solution of these nanocubes were first drop-casted onto one graphene-coated TEM grid. Upon evaporation of the solvent, a small droplet of the acidified etching solution was placed onto the TEM grid followed by encapsulation with another piece of graphene-coated TEM grid to form GLCs.

**AFM measurements**. AFM images were taken by using a MFP-3D AFM (Asylum Research) operating in the tapping mode. GLCs were situated on a glass substrate during AFM scans.

**TEM imaging**. Low-magnification TEM images were acquired on a JEOL JEM 1400plus microscope operating at 120 kV with a LaB₆ filament. HRTEM and STEM-HAADF images of Pd@Au nanocubes were recorded on an aberration-corrected Hitachi HD 2700C STEM microscope equipped with a parallel EELS detector (Gatan Enfina-ER). Samples were prepared using holey 400-mesh Cu grids coated with ultrathin carbon films (Ted Pella). STEM-EDX mapping of Pd@Au nanocubes were performed on a JEOL JEM 3200FS TEM operating at 300 kV. STEM-EELS elemental maps for Cu@Au nanocubes were acquired on an aberration-corrected Hitachi HD 2700C STEM microscope equipped with a parallel EELS detector (Gatan Enfina-ER).

In situ TEM imaging of GLCs was performed on a JEOL JEM 1400plus TEM operating at 120 kV. At least 10 particles were imaged for any given core–shell nanoparticle to ensure that the results presented here are statistically representative. Videos were acquired using a Gatan OneView CMOS camera capable of recording $4k \times 4k$ images at the frame rate of 15 fps controlled by the Gatan in situ video software package. Electron dose rates (unit: electrons/Å² s) were estimated based on the current density $J$ (pA/cm²) measured at the phosphor screen under identical imaging settings for GLCs but without specimen in the electron beam path as

$$\text{Dose rate} = 6.24 \times 10^{-10} \, J(M \cdot 0.77)^2 \qquad (1)$$

where $6.24 \times 10^{-10}$ is the conversion factor between pA/cm² and electrons/Å² s, $M$ is the nominal magnification displayed by the instrument and 0.77 is the post-magnification factor for the camera.

Because dissolution is beam initiated, care was taken to minimize the total electron dose deposited on nanocrystals by navigating GLCs at low magnifications (≤×10,000, electron dose rate: ≤30 electrons/Å² s) with the electron beam spread as much as practically possible. Once a suitable nanocrystal was located, the magnification was increased to either ×100,000 or ×120,000 suitable for single-particle reaction imaging. The electron dose rate was subsequently increased to a desired level by turning the "brightness knob" on the instrument control panel a predetermined number of times, and immediately after that we started TEM movie recording. With the lookback capability of the OneView camera, the start of any dissolution reaction can be readily retrieved.

**Numerical simulation of nanocrystals transformation**. We used a discrete-time lattice-gas MC scheme similar to the one detailed in our prior work to simulate the non-equilibrium transformation of a metal nanocrystal in an oxidative environment[49]. Initial atom positions were chosen on a face-centered cubic lattice according to the geometry and chemical composition of the core@shell microstructure. At every MC step, the state of a randomly chosen site $i$ is switched with

Glauber acceptance probability

$$k_i = 1/\left(1 + \tau e^{-\Delta H_i/k_B T}\right) \tag{2}$$

where $k_B T$ are Boltzmann constant and environmental temperature. A correction factor, which is a function of the number of lattice sites at the surface of the nanocrystal, $\tau$ was used to restore detailed balance because MC moves are restricted to active sites at the nanocrystal surface[19,49]. The energy difference $\Delta H_i$ for the MC step was calculated as the sum of the chemical potential $\mu_i$, the energy $\epsilon_{i,j}$ of the created or broken bonds with the occupied first neighbor sites $j$, and the energy variations $\Delta \epsilon_{j,k,k\neq i}$ of the remaining bonds for site $j$ as

$$\Delta H_i = \pm \left\{ \mu_i + \sum_j \left( \epsilon_{i,j} + \sum_{k\neq i} \Delta \epsilon_{j,k} \right) \right\} \tag{3}$$

Positive (negative) sign was used for atom deposition (removal). We assumed the effective chemical potential is the sum of chemical potential of the oxidative environment $\mu_e$ (applied electrostatic potential), the element-specific reactivity $\mu_0$ (standard electrode potential), and the degree of exposure of the atom site to the liquid in the reactor cell, $\mu = \mu_e + \mu_0 + \delta_\mu$. Here, $\delta_\mu = \mu^* \tilde{\rho}^m$ with relative solid atom concentration $\tilde{\rho} = \rho/\rho_{bulk}$ is a penalty factor that accounts for the decrease of the concentration of the oxidative ions due to both the local porosity and the graphene blocking effect. We computed the solid atom concentration $\rho$ by summation of Gaussian weights for the distances from the site of interest to other occupied lattice sites in the system. Graphene sheets were described as a set of occupied, yet non-reactive lattice sites. Any change of occupation state was forbidden. Parameters were calibrated based on experimental observations to be $\mu^* = 0.20$ eV and $m = 6.0$. The standard deviation of the Gaussian weight was set equal to 5/4 of the unit cell parameter.

**Estimation of the atom bond energy**. We measured the atom bond energy as a function of local coordination (Supplementary Fig. 13) using classical molecular dynamics (MD) simulations performed with the LAMMPS software[55]. Cu, Pd, and Au monometallic nanocrystals with different shapes were equilibrated at room temperature using a Nosè–Hoover thermostat with 1 fs time integration in consecutive NVE and NVT ensembles[56,57]. Interactions were described using embedded atom method (EAM) and Sheng pair interaction potentials[58]. After reaching a steady-state system equilibrium, a sequence of 500 snapshots were sampled at 2 ps constant interval time from a microcanonical NVE ensemble simulation. The time-averaged per-atom potential energy was then computed. Although atoms are either on the flat surfaces or at the edges or corners of cubic, rhombic-dodecahedron, and octahedron particles, small potential energy differences were measured for those with the same CN, defined as the number $\eta$ of first neighbors. We assumed that the time-averaged per-atom potential energy $E(\eta)$ is evenly distributed among all $\eta$ neighbor bond pairs such that $\epsilon(\eta) = E(\eta)/2\eta$. The atom bond energy between atoms of different chemical element was approximated by the geometric average of the corresponding pure pairs. To improve the performance of the MC algorithm and extrapolate the atom bond energy for low coordinated atoms, we parametrized the MD calculated values with the functional

$$\epsilon(\eta) = \frac{A_1 - A_2}{1 + (\eta/\eta_0)^P} + A_1 + \sigma \tag{4}$$

where $\sigma$ is a Gaussian noise contribution. Parameters for the pure elements and their binary alloys are listed in Supplementary Table 1.

**MC relative reaction time**. We defined a relative time $t_0$ based on the assumption that the diameter of a spherical crystal exposed to a strong oxidative environment decreases at a constant rate. The number of atoms depositing or detaching to the nanocrystal is therefore proportional to its surface area. The interval of relative time between two accepted MC moves was calculated as the sum of the ratios between the numbers of attempted atom removal (or deposition) steps over the number of occupied surface sites (or their non-occupied neighbors). Relative times were scaled to match experiments in the last four frames of Figs. 3, 5, 6 and the central three frames of Supplementary Fig. 10.

## Data availability

All necessary data generated or analyzed during this study are included in this published article, and other auxiliary data are available from the corresponding authors upon request.

## Code availability

The Matlab scripts necessary to reproduce our results are available from the corresponding author (X.Y.) upon request.

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

## Acknowledgements

L.C., J.C., M.C., and X.Y. acknowledge support by the start-up fund from Indiana University. We thank Indiana University Nanoscale Characterization Facility and Electron Microscopy Center for access of instrumentation. This work made use of facilities at Center for Functional Nanomaterials, which is a U.S. DOE Office of Science Facility at Brookhaven National Laboratory (Contract no. DE-SC0012704). L.C. acknowledges a graduate student scholarship from China Scholarship Council. A.L. and M.E. acknowledge funding from Deutsch Foschungsgemeinschaft through the Cluster of Excellence Engineering of Advanced Materials (EXC 315/2), as well as support from the Central Institute for Scientific Computing (ZISC) and the Inter-disciplinary Center for Functional Particle Systems (FPS) at Friedrich-Alexander University Erlangen-Nürnberg. Computational resources and support provided by Erlangen Regional Computing Center (RRZE) are gratefully acknowledged. This work made use of facilities at Center for Functional Nanomaterials, which is a US DOE Office of Science Facility, at Brookhaven National Laboratory under Contract No. DE-SC0012704.

## Author contributions

X.Y. conceived the idea and supervised the project. L.C. and M.C. synthesized nano-crystals. L.C. performed TEM experiments. J.C. contributed to data analysis. A.L. developed computational algorithms and performed simulations. N.L. and D.S. con-tributed to HRTEM and STEM-EDX characterization. L.C., A.L., Q.Z., M.E., and X.Y. co-wrote the manuscript. M.E. supervised the computational contribution to the project. All authors discussed the results and commented on the manuscript.

## Competing interests

The authors declare no competing interests.
