## [Peer Review File · Nature Communications]

Reviewers' comments:

Reviewer #1 (Remarks to the Author):

In this manuscript, the authors investigated the reaction kinetics during the dissolution of well-defined bimetallic core-shell nanocubes using real-time graphene-liquid-cell (GLC) TEM imaging and Monte Carlo simulations. They discovered the anisotropy reaction environment within GLCs and the slowdown in the Au shell etching by conducting control experiment and careful data analysis. Overall, this appears to an interesting work in which authors used advanced electron microscopy and simulations are used, and manuscript is well organized and written. This work can be accepted after addressing the following questions:

1. How did the authors control the galvanic replacement between palladium and gold in such controllable way since the as-synthesized core-shell nanocubes show very uniform and distinct boundary?
2. The author claimed the kinetics of the etching can be referred to the etching behavior, how did the author get this conclusion? Is there any derivation process should be included in this manuscript?
3. How did the authors measure the volume of gold shell after the palladium shell was exposed, since the observed islands of gold were not uniform?
4. In Figure 5, the author claimed that the slowdown effect came from the electron transfer from palladium to gold. However, the author also claimed that the electrons can be transferred from gold region to palladium region.
5. For the etching of Cu@Au system, it should have the slowdown effect as well during the etching process. Did the authors observe the similar phenomenon?
6. The authors claimed that the etching of Cu@Au system could be a method for measuring shell thickness and uniformity of particles. Can this in situ method be easily conducted with high reproducibility?
7. It seems that the core is not really visible in the time-dependent TEM images, how the authors make sure it is indeed co-shell particle? To me, some of the 3D models given are not really comparable to 2D TEM images. For example, see 5a, 6(b,c), 6d. Is it possible look at the final particle obtained after etching by STEM?
8. Can authors provide any statistics on the number of particles they looked at for the final conclusions?
9. Is there any comparison between in situ TEM and in solution phase synthesis in a flask? Can authors translate this understanding to prepare a certain shape in large scale? I mean how this study lead to something useful for synthesis of shapes shown in 3D models in a flask?

Reviewer #2 (Remarks to the Author):

This is a joint experimental/simulation paper about the dissolution of core-shell nanocrystals - the shapes that occur and the mechanisms by which dissolution occurs. I do think this is interesting. However, I feel uncertain this work is sufficient for a high-impact journal for several reasons.

1. I feel the manuscript is written in a misleading way in several places. First, when I read the abstract, I got the impression that most of the theoretical work was done by molecular dynamics. However, upon reading the paper I found that most of the theory was done with Monte Carlo simulations. At the bottom of p. 9 (lines 294-296) they say "Our observations revealed diverse dissolution reaction pathways that provide access to bimetallic multipods, segmental nanorods, nanoboxes and nanocages via controlled etching of core-shell nanocubes." However, when I searched through the paper, this sentence was the only place (other than the references) where "multipod", and "nanocage" were found. The paper should be cleaned up to more accurately reflect the work that was done.

2. A significant conclusion from this work is that the graphene cell used in this work significantly influences the etching kinetics. While I feel it is valuable for others to know this, it also limits the applicability of the results to the anisotropic system studied here. It would not apply to general dissolution outside of the graphene cell. Because of this limited scope, I feel the work is not broadly significant and it should appear in a more specialized journal.

3. In lines 454-456, the authors state that τ is a correction factor to restore detailed balance. I was trying to imagine why their simulations might not satisfy detailed balance and I guess this is because of the way they obtain ϵ - this should be clarified. It is also unclear to me how just one constant factor can address every instance of a transition not satisfying detailed balance. This definitely needs to be clarified.

4. I feel their whole procedure for obtaining $\epsilon(\eta)$ is suspect. Why don't they enumerate the "before" and "after" local configurations as a crystal dissolves and get the change in energy that way? Better than this they should kinetic Monte Carlo based on DFT calculations. I feel that their model is very qualitative, parameters are fit to the experiments, and it could match any experimental observation they choose.

Reviewer #3 (Remarks to the Author):

This paper by Lei Chen et al reports findings on the chemical etching kinetics of bimetallic core-shell nanoparticles uncovered using in-situ liquid cell TEM. In addition to previously reported behavior from similar studies on monometallic particles, the authors experimentally observe a marked slowing in dissolution rate for Pd@Au nanocubes at the point both metals are exposed, both for the particle overall and for the less-reactive Au metal. They further demonstrate the significance of metal reactivity on dissolution kinetics by rapidly etching the Cu core of Cu@Au nanocubes. The authors also find that sides of the particle likely in contact or in close proximity to the enveloping graphene sheets of the liquid

cell are etched much more slowly due to limited exposure to the oxidative environment. Lastly, they support the majority of their experimental findings with agreeing Monte Carlo simulation results.

I believe the paper is appropriate for publication, with a few potential revisions. These findings are significant in building on the rapidly expanding field of in-situ liquid cell TEM as well as multimetallic nanoparticle chemistry by combining the two. Namely, they extend previous liquid cell experimental frameworks to a more complex material system involving multiple reactive metal species. While the findings from their model nanocube system are interesting in their own right, I believe the main contribution is the demonstration that our current liquid cell TEM techniques can capture the chemistry of more catalytically-relevant (for instance) materials with sufficient temporal and spatial resolution.

In my view of the recorded in-situ videos and subsequent analysis, with results corroborated by simulations, I see no glaring issue with the authors' conclusions and agree with their major findings. That being said, I suggest the following critical feedback be taken into consideration for the final manuscript:

1. Throughout the "Intrinsic and extrinsic factors responsible for anisotropic dissolution" section, when discussing the orientation of the Pd@Au nanocube relative to the confining graphene windows, the language is unclear regarding what described directions refer to what face of the original cube. In the first paragraph, "top and bottom" are both defined with respect to Fig. 3a after rotation as well as with respect to the initial cube orientation orthogonal to the electron beam. In subsequent paragraphs, which "top and bottom" facets are being referred to is not explicitly clear, especially to readers less familiar with the sealing behavior of graphene liquid cells. I suggest defining an e-beam axis and defining orientations relative to that.

2. I think the absence of anisotropic etching in many other GLC-confined experiments is worth exploring further past the putative statements given—it is strange that monometallic etching in both this and previous papers had not observed that behavior, and I don't believe it is fully explained by the smaller particle size (other than chance with small sample size) since GLC pockets are subject to large variations in thickness and in the reference cited are likely to "wrap" even smaller particles than the ones studied. Rather, the later remark that there were simply a range of liquid thicknesses present and some happened to wrap on particles (more likely the larger ones) could be verified by measurements of the liquid cell thickness for both cases where the anisotropic etching is and isn't observed. Such a measurement before and after etching could also verify whether or not the pocket thickness is defined by the particle size in certain size regimes.

3. It was not immediately obvious to me the purpose of investigating the etching of cubes with thinner Au shells (as opposed to simply gathering more statistics with the same shell thickness). What was the expected effect of a larger Pd:Au ratio?

4. Please include experimental details of the AFM measurements in the methods.

5. I would include a STEM-EDX map of the Cu@Au nanocages similar to that of Fig. 1 to verify the elemental distribution (placed in the SI).

6. The Cu@Au nanocube experiments seems overall disconnected from the larger picture. The abstract implies they experience similar behavior to Pd@Au nanocubes, while the discussion/conclusion section disregards them entirely. In the earlier kinetics analysis, the lack of a linear V vs. t relationship already ruled out rate limiting from the oxidative species—was this section just to verify this? Were there similar observations regarding galvanic corrosion? This section could be motivated better to be less of an afterthought, especially where it lies near the conclusion of the paper.

7. In Fig. 4, it may help to mark the expected nanocube locations in 4d (or S6), since the bright spots in the AFM don't match up exactly with the cubes are located in the correlated SEM image. (Also, is this due to particle movement? Even though in a month the cubes are barely seen to move.)

Reviewer #1 (Remarks to the Author):

In this manuscript, the authors investigated the reaction kinetics during the dissolution of well-defined bimetallic core-shell nanocubes using real-time graphene-liquid-cell (GLC) TEM imaging and Monte Carlo simulations. They discovered the anisotropy reaction environment within GLCs and the slowdown in the Au shell etching by conducting control experiment and careful data analysis. Overall, this appears to an interesting work in which authors used advanced electron microscopy and simulations are used, and manuscript is well organized and written. This work can be accepted after addressing the following questions:

Our response: We thank the reviewer for the positive comments.

1.1. How did the authors control the galvanic replacement between palladium and gold in such controllable way since the as-synthesized core-shell nanocubes show very uniform and distinct boundary?

Our response: Synthesis of Pd@Au core-shell nanocubes was based on a literature report¹ with an important modification. Specifically, we lowered the reaction temperature from 95 °C, the value reported previously, to 25 °C such that alloying between Pd and Au at the core-shell interfaces is further suppressed.

Galvanic replacement between Pd(0) and Au(I) is not thermodynamically favorable due to the following reason: Galvanic displacement takes place when the base metal that is displaced by metallic ions in solution has a more negative electrode potential than the displaced metal ions. The Au(III) precursor is believed to be immediately reduced to Au(I) upon addition to the reaction mixture due to the presence of L-ascorbic acid (standard reduction potential of L-ascorbic acid: -0.081V vs. normal hydrogen electrode (NHE))² employed in our synthesis. With the standard potentials $E^0(\text{Pd}^{2+} / \text{Pd}) = 0.915\text{ V}^3$ and $E^0([\text{AuBr}_2]^- / \text{Au}) = 0.96\text{ V}$,⁴ we have

The thermodynamic driving force for galvanic replacement between Pd(0) by $[\text{AuBr}_2]^-$ ions is as small as 45 meV under standard conditions (requiring all components at unit activity). When considering the low concentration of $[\text{AuBr}_2]^-$ species in reaction solution (0.05 mM, rather than 1.0 M under standard conditions), the driving force for galvanic replacement is essentially negligible.

1.2. The author claimed the kinetics of the etching can be referred to the etching behavior, how did the author get this conclusion? Is there any derivation process should be included in this manuscript?

Our response: The correlation between kinetics of etching and etching behavior was established within the framework of the Lifshitz, Slyozov and Wagner (LSW) theory originally developed to describe crystal growth.^{5,6} A recent study on dissolution of mono-metallic nanoparticles also provided derivations for the relationship between dissolution

kinetics and etching behavior,⁷ and we refer the reviewer to the relevant literature for details. It is worth noting that in Ref. 7, the etching behavior of monometallic Pt nanoparticles was only deduced by performing fitting to the etching kinetics, whereas our work directly captured detailed etching behavior (e.g., edge-selective atom removal vs. layer-by-layer atom removal) of bimetallic nanoparticles (see for example Fig. 3d in the main text), in addition to kinetic analysis.

1.3. How did the authors measure the volume of gold shell after the palladium shell was exposed, since the observed islands of gold were not uniform?

Our response: After the Pd shell was exposed, variations in volumes of residual Au islands were observed for a single core-shell nanocube. In particular, the two facets initially close to the graphene s were etched at a slower rate. We therefore only analyzed those islands that were not in close proximity with the graphene windows throughout the etching process. Their volume was measured individually and averaged. The total volume of the gold shell after the palladium shell was exposed was calculated as six times this average island volume. This calculation returns the volume of the gold shell ignoring the graphene blocking effect.

1.4. In Figure 5, the author claimed that the slowdown effect came from the electron transfer from palladium to gold. However, the author also claimed that the electrons can be transferred from gold region to palladium region.

Our response: The negative charge of electrons released at individual oxidation events spreads across the two regions palladium and gold. This can occur in principle by electron transfer in either direction. Whereas the equilibrium net flux is from palladium to gold due to the higher standard electrode potential of gold as indicated by the thick arrow in Fig. 5, the reverse direction as indicated by the dashed arrow in Fig. 5 is also possible momentarily.

1.5. For the etching of Cu@Au system, it should have the slowdown effect as well during the etching process. Did the authors observe the similar phenomenon?

Our response: The slowdown of Au shell etching is due to the cathode protection mechanism. Specifically, the more reactive metal in the core, when exposed to the etching solution, becomes the sacrificial anode and dissolve preferentially protecting the galvanic cathode (i.e., the Au shell). We believe that the slowdown effect remains valid for the Cu@Au system, yet the timescale over which it was operative is determined by how fast Cu gets etched. Since the Cu core was etched away completely within ca. 60 ms under typical *in-situ* TEM reaction conditions and the time resolution of our imaging camera is 40 ms, it is extremely difficult to observe experimentally the slowdown effect induced by Cu.

1.6. The authors claimed that the etching of Cu@Au system could be a method for measuring shell thickness and uniformity of particles. Can this in situ method be easily conducted with high reproducibility?

Our response: We thank the reviewer for this thoughtful question. In the manuscript, we made the statement that “a potential application of GLC TEM imaging of particle etching could be measuring shell thickness and uniformity within an ensemble of core-shell

nanoparticles by imaging the dissolution behavior of multiple objects in the same field of view...” We view it as *a forward-looking statement rather than a claim substantiated by existing evidence*. We do believe that this *in-situ* method can be conducted with high reproducibility, provided that a number of important experimental parameters precisely controlled to ensure reproducible conditions. For example, reproducible control of the liquid environment inside GLCs and electron beam current and illumination condition are critical to observing predictable dissolution behavior. To this end, we are encouraged to see that some of these challenges are already being addressed by researchers within the *in-situ* TEM community. For instance, a recent work by the Alivisatos group reported the development of Digital Micrograph script to calibrate and to control the condenser system of TEM, allows the electron dose rate to be modulated reproducibly such that the effects of the dose rate on the etching rate of monometallic Au nanoparticles can be studied.⁸

1.7. It seems that the core is not really visible in the time-dependent TEM images, how the authors make sure it is indeed core-shell particle? To me, some of the 3D models given are not really comparable to 2D TEM images. For example, see 5a, 6(b,c), 6d.

Our response: We are fairly certain that the particles imaged under *in-situ* TEM are core-shell ones because according to STEM-HAADF imaging results (e.g., Fig. 1 for Pd@Au, Fig. 6a for Cu@Au), the morphological yield of core-shell nanoparticles were basically 100% in our synthesis. The 3D structural models of transient intermediates were snapshots extracted from atomistic simulations. To synchronize experimental and simulation movies, we strived to locate precisely featured frames such as the one when core metal begins to get exposed to the solution environment, as illustrated in **Figure R1** below. The different contrast between central and peripheral regions of the nanocube at the later stage of dissolution reaction further confirms that they were indeed core-shell particles. Such contrast difference would have been absent during dissolution of monometallic nanoparticles.⁸⁻¹⁰

Figure R1. TEM movie frames extracted from Movies (a) S2, (b) S7a and (c) S7b showing the moment when core metal starts to get exposed to the etching solution. The dotted red ellipses highlight discontinuity in contrast under bright-field TEM imaging conditions.

Is it possible to look at the final particle obtained after etching by STEM?

Our response: Because particle etching is initiated and sustained primarily through secondary species generated in solution upon electron beam illumination, it remains very

difficult, if not impossible, to halt the reaction completely and image “the final particle obtained after etching” in STEM mode.

1.8. Can authors provide any statistics on the number of particles they looked at for the final conclusions?

Our response: We imaged a minimum of ten particles for any given core-shell nanoparticle to ensure that the results presented in the manuscript are statistically representative for that particular nanoparticle (e.g, Pd@Au nanocube with 44-nm Pd core and 35-nm Au shell). Following this reviewer’s suggestion, we added the following statement in the Method section of the revised manuscript: “**At least ten particles were imaged for any given core-shell nanoparticle to ensure that the results presented here are statistically representative.**”

1.9. Is there any comparison between in situ TEM and in solution phase synthesis in a flask? Can authors translate this understanding to prepare a certain shape in large scale? I mean how this study lead to something useful for synthesis of shapes shown in 3D models in a flask?

Our response: We acknowledge that the oxidative condition used in the present work is different from what’s typically employed in the *ex-situ* solution-phase etching reaction, involving both the FeCl₃ etchant and oxidative radicals generated by the electron beam. In a bulk synthesis, stirring and mixing is important on that macroscopic scale to ensure a homogeneous reaction environment, while for the *in-situ* single-particle study, good mixing only needs to occur on the nanometer scale. Because the time scale of the *in-situ* etching reaction is on the order of a few seconds or longer, and the particle does not move significantly throughout the reaction, the diffusion of species in solution surrounding the particle is extremely rapid over the same time periods, creating a well-mixed system. Therefore, we believe that mixing should not be a concern for *in-situ* etching reactions.

Furthermore, our atomistic simulations reproduce experimental shape transformations extremely well. These results clearly show that although there are differences in the specific oxidation chemistries between our *in-situ* and bulk *ex-situ* etchings, the single-particle reaction trajectories are unlikely to be altered by the *in-situ* reaction conditions. We do believe that the knowledge gained from the present *in-situ* study can be extended to the more general bulk system. It is beyond the scope of this work to replicate the non-equilibrium reaction pathways in solution phase synthesis but such work should be performed in future.

Reviewer #2 (Remarks to the Author):

This is a joint experimental/simulation paper about the dissolution of core-shell nanocrystals - the shapes that occur and the mechanisms by which dissolution occurs. I do think this is interesting. However, I feel uncertain this work is sufficient for a high-impact journal for several reasons.

Our response: We thank the referee for finding our work interesting. We believe that our work is suitable for a high-impact journal such as *Nature Communications* because of the ground-breaking nature of the characterization techniques involved that capture atomic-level mechanism of non-equilibrium shape transformations of core-shell nanocrystals and deducing the local physicochemical conditions inside TEM liquid cells for the first time. Our work is quantitative, shows good agreement between atomistic simulation and experiment, and is of interest to a broad audience including materials scientists and electrochemists.

2.1. I feel the manuscript is written in a misleading way in several places. First, when I read the abstract, I got the impression that most of the theoretical work was done by molecular dynamics. However, upon reading the paper I found that most of the theory was done with Monte Carlo simulations.

Our response: We agree with the referee. Both methods (molecular dynamics and Monte Carlo) are used in the paper, with Monte Carlo being more prominent. The correct term should therefore be “**atomistic simulation**” instead of “molecular dynamics” as this term encompasses both techniques. We corrected that term in the abstract of the manuscript.

2.2. At the bottom of p. 9 (lines 294-296) they say "Our observations revealed diverse dissolution reaction pathways that provide access to bimetallic multipods, segmental nanorods, nanoboxes and nanocages via controlled etching of core-shell nanocubes." However, when I searched through the paper, this sentence was the only place (other than the references) where "multipod", and "nanocage" were found. The paper should be cleaned up to more accurately reflect the work that was done.

Our response: Indeed, the terminology had not been used consistently. But all work mentioned in the text was done as described. To this end, we made two revisions. (1) The text “[.] **a bimetallic multipod was observed after 137s.**” has been added to the description of Figure 3a on page 5. (2) The terms “nanobox” and “nanocage” both refer to a nanoscale object with a hollow interior, yet a “nanocage” is more porous than a “nanobox”. The original statement on page 9, “Two levels of contrast were discerned between the time period of 4.2-5.6 s and after 5.6 s, suggesting that opposing facets of the Au nanobox dissolved sequentially (Fig. 6d, Movie S10a).”, now reads “Two levels of contrast were discerned between the time period of 4.2-5.6 s and after 5.6 s, suggesting that opposing facets of the Au nanobox dissolved sequentially **forming a nanocage with enhanced porosity** (Fig. 6d, Movie S10a).”

2.3. A significant conclusion from this work is that the graphene cell used in this work significantly influences the etching kinetics. While I feel it is valuable for others to know this, it also limits the applicability of the results to the anisotropic system studied here. It would

not apply to general dissolution outside of the graphene cell. Because of this limited scope, I feel the work is not broadly significant and it should appear in a more specialized journal.

Our response: We agree with the reviewer that one of the major conclusions from this work is that the graphene windows can perturb the kinetics and pathways of reactions under study. Recognizing the rapidly growing interests in *in-situ* liquid cell electron microscopy, we believe that a clear demonstration and interpretation on how the electron transparent membrane, whether it is Si₃N₄, graphene or MoS₂, may alter the kinetic processes under study using well-defined model systems is of general interest and thus appealing to the broad chemistry and materials community that *Nature Communications* reaches. We expect that once published, this manuscript will: 1) raise the bar for data quality and proper interpretation of upcoming *in-situ* liquid cell TEM works, and 2) inspire innovations to overcome this challenge such that the nanoscale objects being studied no longer play the role of “spacer particles”.

More importantly, we do believe that our paper *goes far beyond* the idea of graphene membranes can influence etching kinetics. We have performed detailed analysis of the core-shell particle dissolution profile and have conducted extensive atomistic simulations of single-particle etching reactions. These efforts provide new insights into both the intrinsic and extrinsic factors governing the anisotropic dissolution kinetics. The coherence between our experimental data and simulation results convey to the readers insights with predictive power on the important links between microscopic atomic processes and macroscopic shape transformations. As commented by Reviewer 3, “*these findings are significant in building on the rapidly expanding field of in-situ liquid cell TEM as well as multimetallic nanoparticle chemistry by combining the two.... While the findings from their model nanocube system are interesting in their own right, I believe the main contribution is the demonstration that our current liquid cell TEM techniques can capture the chemistry of more catalytically-relevant (for instance) materials with sufficient temporal and spatial resolution.*”

2.4. In lines 454-456, the authors state that τ is a correction factor to restore detailed balance. I was trying to imagine why their simulations might not satisfy detailed balance and I guess this is because of the way they obtain ϵ - this should be clarified. It is also unclear to me how just one constant factor can address every instance of a transition not satisfying detailed balance. This definitely needs to be clarified.

Our response: We thank the reviewer for the suggestion of clarification. The factor τ corrects for the specific choice of Monte Carlo moves and results from a simple counting argument. The same factor was introduced in the prior work *Science* (2016) and used in *ACS Nano* (2018) (References 19 and 49 in the manuscript). These references contain a complete derivation of the general formula used in the present work.

In brief: Detail balance is violated because MC moves are restricted to sites where each type of moves is plausible: insertion at empty sites and deletion of occupied sites. Only atoms at the surface of the nanocrystal can be deleted, and atoms can be inserted only at sites directly connected to occupied sites. At any time, there are more empty than occupied sites at the surface of the nanocrystal. Given that the MC algorithm evenly attempts insertion or deletion, insertion to any empty site has lower probability to be attempted than deletion of any

occupied site. To recover detail balance, the probability of attempting moves has to be corrected. This is done by introducing the τ factor, which depends on the number of lattice sites at the surface of the nanocrystal.

We revised the statement in lines 454-456 of the Methods section, which now reads: “A correction factor τ , which is a function of the number of lattice sites at the surface of the nanocrystal, was used to restore detailed balance because MC moves are restricted to active sites at the nanocrystal surface.^{19,49}”.

2.5. I feel their whole procedure for obtaining $\varepsilon(\eta)$ is suspect. Why don't they enumerate the "before" and "after" local configurations as a crystal dissolves and get the change in energy that way? Better than this they should kinetic Monte Carlo based on DFT calculations. I feel that their model is very qualitative, parameters are fit to the experiments, and it could match any experimental observation they choose.

Our response: Assuming the bond energy ε is a function of local coordination η is a significant improvement from the previous variant of the model introduced in Science (2016) and extended in ACS-Nano (2018), which used a constant bond energy estimated from the sublimation energy of a pure bulk phase. The computation of the energy difference between the system configurations “before” and “after” an attempted move proposed by the referee is indeed what we already do in the present method. The use of DFT calculations would not lead to more reliable bond energy parameters compared to classical MD calculations. After all, EAM pair interaction potentials are calibrated to reproduce DFT calculations. Furthermore, we believe the approximations made by using MC in the first place is more significant than the difference between EAM and DFT. However, given the scope of calculations necessary in this work (many millions of atoms), a more accurate calculation is currently out of reach. We would like to stress that the procedure for obtaining $\varepsilon(\eta)$ is not a fit to our experiment. The only free parameters derived from our experimental data are the time scaling factor and the oxidative chemical potential. In light of this fact, we feel the quantitative predictions made with our simulations (Fig. 2-6) fit the liquid cell observations remarkably well.

Reviewer #3 (Remarks to the Author):

This paper by Lei Chen et al reports findings on the chemical etching kinetics of bimetallic core-shell nanoparticles uncovered using in-situ liquid cell TEM. In addition to previously reported behavior from similar studies on monometallic particles, the authors experimentally observe a marked slowing in dissolution rate for Pd@Au nanocubes at the point both metals are exposed, both for the particle overall and for the less-reactive Au metal. They further demonstrate the significance of metal reactivity on dissolution kinetics by rapidly etching the Cu core of Cu@Au nanocubes. The authors also find that sides of the particle likely in contact or in close proximity to the enveloping graphene sheets of the liquid cell are etched much more slowly due to limited exposure to the oxidative environment. Lastly, they support the majority of their experimental findings with agreeing Monte Carlo simulation results. I believe the paper is appropriate for publication, with a few potential revisions. These findings are significant in building on the rapidly expanding field of in-situ liquid cell TEM as well as multimetallic nanoparticle chemistry by combining the two. Namely, they extend previous liquid cell experimental frameworks to a more complex material system involving multiple reactive metal species. While the findings from their model nanocube system are interesting in their own right, I believe the main contribution is the demonstration that our current liquid cell TEM techniques can capture the chemistry of more catalytically-relevant (for instance) materials with sufficient temporal and spatial resolution.

In my view of the recorded in-situ videos and subsequent analysis, with results corroborated by simulations, I see no glaring issue with the authors' conclusions and agree with their major findings. That being said, I suggest the following critical feedback be taken into consideration for the final manuscript:

Our response: We thank the reviewer for the positive comments and enthusiasm about our work.

3.1. Throughout the “Intrinsic and extrinsic factors responsible for anisotropic dissolution” section, when discussing the orientation of the Pd@Au nanocube relative to the confining graphene windows, the language is unclear regarding what described directions refer to what face of the original cube. In the first paragraph, “top and bottom” are both defined with respect to Fig. 3a after rotation as well as with respect to the initial cube orientation orthogonal to the electron beam. In subsequent paragraphs, which “top and bottom” facets are being referred to is not explicitly clear, especially to readers less familiar with the sealing behavior of graphene liquid cells. I suggest defining an e-beam axis and defining orientations relative to that.

Our response: We very much appreciate this suggestion. We agree with the reviewer that the phrase “top and bottom facets” might have been confusing at several places. To remove any ambiguity concerning our descriptions, the following revisions has been made:

- 1) The phrase “top and bottom” is now reserved for the two nanocube facets that are orthogonal to the electron beam before etching starts. In other words, “top and bottom” always refers to the two facets that are in close proximity to the graphene

windows at the beginning of reaction. In maintext where “top and bottom” was first mentioned, we have added the following sentence: “**In the rest of this paper, “top and bottom” facets refers to the two nanocube facets that are orthogonal to the electron beam before etching starts.**”

- 2) At places where particle rotation occurred upon etching, the phase “top and bottom” has been replaced with “**upper and lower**” to refer to domains/islands on the specific TEM image.

3.2. I think the absence of anisotropic etching in many other GLC-confined experiments is worth exploring further past the putative statements given—it is strange that monometallic etching in both this and previous papers had not observed that behavior, and I don't believe it is fully explained by the smaller particle size (other than chance with small sample size) since GLC pockets are subject to large variations in thickness and in the reference cited are likely to “wrap” even smaller particles than the ones studied. Rather, the later remark that there were simply a range of liquid thicknesses present and some happened to wrap on particles (more likely the larger ones) could be verified by measurements of the liquid cell thickness for both cases where the anisotropic etching is and isn't observed. Such a measurement before and after etching could also verify whether or not the pocket thickness is defined by the particle size in certain size regimes.

Our response: We thank the reviewer for these insightful comments, especially the proposed experiments of determining the liquid cell thickness for both scenarios where anisotropic etching is and is not observed. The truth is this *graphene-induced anisotropic etching was observed in almost every single movie we acquired for this project*. Due to variations in the size of core-shell nanocube, tightness of graphene wrapping and etching kinetics, this “extrinsic” anisotropic etching was more obvious for some reactions (e.g., Movies S2a, S7a, S7b) while being less apparent in others (e.g., Movies S1a, S6, S8). We suspect that this graphene-induced anisotropic etching is also present in the dissolution reaction of *monometallic nanoparticles*, yet it would be extremely difficult to recognize such reaction anisotropy due to lack of contrast across different areas on a nanoparticle under typical bright-field imaging conditions. To better illustrate this point, we have compiled **Figure R2**, where the left column are movie frames extracted from different Pd@Au etching reactions, whereas the right column are corresponding images generated by applying the grayscale pixel intensity extracted from the central part of the nanoparticle to all other pixels of that particle by using Photoshop. Clearly, differences in the etching behavior between the “top and bottom” nanocube facets vs. their side facets would have been “lost”. Accordingly, we argue in the manuscript that “*Our work demonstrates that bimetallic core-shell nanocrystals are excellent probes for the local physicochemical conditions inside TEM liquid cells.*”

Figure R2. Left column: movie frames extracted from different Pd@Au etching reactions. Distinguishing features suggesting anisotropic dissolution of the Au shell are marked with white irregular circles on each image. Right column: corresponding images generated by applying a common pixel intensity value to all pixels belonging to the nanoparticle by using Photoshop.

3.3. It was not immediately obvious to me the purpose of investigating the etching of cubes with thinner Au shells (as opposed to simply gathering more statistics with the same shell thickness). What was the expected effect of a larger Pd:Au ratio?

Our response: The main purpose of studying the etching of nanocubes with a larger Pd:Au volume ratio is to better visualize and to quantify the slowdown in etching kinetics of Au shell. Starting from Pd@Au nanocubes with thinner Au shells coupled with a strong oxidizing environment, the slowdown in Au shell dissolution would be more obvious because the Au shell would have been etched away rapidly if the cathodic protection due to the Pd domain were not operative.

3.4. Please include experimental details of the AFM measurements in the methods.

Our response: We thank the reviewer for catching this oversight. Experimental details of the AFM measurements are now included in the Methods section.

3.5. I would include a STEM-EDX map of the Cu@Au nanocages similar to that of Fig. 1 to verify the elemental distribution (placed in the SI).

Our response: We thank the reviewer for this suggestion. During the round of manuscript revision, we have carried out extensive GLC TEM experiments in order to obtain STEM-EDX map of Au nanocages resulting from etching of Cu@Au nanocubes. The major challenge lies in the fact that *in-situ* GLC particle etching is initiated and sustained primarily through secondary species generated in solution upon electron beam illumination. Consequently, it is very difficult, if not impossible, to halt the etching reaction completely and image the particle in STEM mode. Fortunately, several Au nanocages were found at

GLCs that became dried out (likely due to smaller than normal volume of some GLC pockets) shortly after etching started, making it suitable for subsequent STEM-EDX mapping as further etching during STEM scans is inhibited). In the revised manuscript, representative STEM-EDX mapping data of Au nanocages have been included in the new **Supplementary Figure 11**, which is also shown below.

(New) Supplementary Figure 11 | TEM characterization of Cu@Au nanocubes. (a) TEM image of Cu nanocubes (edge length: 44.3 ± 3.1 nm). (b) TEM image of Cu@Au nanocubes (shell thickness: 2.6 ± 0.4 nm). (c) STEM-EDX elemental mapping of Au nanocages resulting from etching of Cu@Au nanocubes. The GLCs became dried out shortly after etching started, making it suitable for subsequent STEM-EDX mapping as further etching during STEM scans is inhibited. The Cu signal observed on STEM-EDX maps is likely attributed to precipitated Cu salts as a result of sample dryout.

3.6. The Cu@Au nanocube experiments seems overall disconnected from the larger picture. The abstract implies they experience similar behavior to Pd@Au nanocubes, while the discussion/conclusion section disregards them entirely. In the earlier kinetics analysis, the lack of a linear V vs. t relationship already ruled out rate limiting from the oxidative species—was this section just to verify this? Were there similar observations regarding galvanic corrosion? This section could be motivated better to be less of an afterthought, especially where it lies near the conclusion of the paper.

Our response: The main purpose of studying dissolution of Cu@Au nanocubes in addition to the Pd@Au systems is to understand how strongly dissimilar reactivity between core and shell metals influence reaction pathways. Distinct from dissolution of Pd@Au nanocubes where core and shell metals remain physically connected during most of the reaction time, the more reactive Cu core gets etched away almost instantaneously, yielding an Au nanobox that became an Au nanocage upon further etching. We argue that the Pd@Au and Cu@Au provide two distinct types of reaction pathways for dissolution of bimetallic core-shell nanocubes. Both systems are important for understanding sculpting at the nanoscale and provide some predictive power for non-equilibrium nanocrystal shape transformations.

3.7. In Fig. 4, it may help to mark the expected nanocube locations in 4d (or S6), since the bright spots in the AFM don't match up exactly with the cubes are located in the correlated SEM image. (Also, is this due to particle movement? Even though in a month the cubes are barely seen to move.)

Our response: Following this reviewer's suggestion, we have added markers (blue rectangular boxes) on the SEM and AFM images of Figure S6a to help readers better visualize the expected nanocube locations. The "mismatch" between bright spots on AFM and SEM scans for a given sample is likely due to the two different imaging modes. However, movement of particles, albeit to different extents for different particles, was obvious by comparing SEM images recorded from the same sample area on the day the GLC was made versus after one month, further confirming the presence and the integrity of liquid pockets.

References

- 1 Lim, B. *et al.* Synthesis of Pd–Au bimetallic nanocrystals via controlled overgrowth. *J. Am. Chem. Soc.* **132**, 2506-2507 (2010).
- 2 Fruton, J. Oxidation-reduction potentials of ascorbic acid. *J. Biol. Chem.* **105**, 79-85 (1934).
- 3 Bard, A. J. & Faulkner, L. R. Fundamentals and applications. *Electrochemical Methods* **2**, 580-632 (2001).
- 4 Evans, D. H. & Lingane, J. J. Standard potentials of the couples involving AuBr_4^- , AuBr_2^- and Au in bromide media. *J. Electroanal. Chem.* **6**, 1-10 (1963).
- 5 Lifshitz, I. M. & Slyozov, V. V. The kinetics of precipitation from supersaturated solid solutions. *J. Phys. Chem. Solids* **19**, 35-50 (1961).
- 6 Viswanatha, R. & Sarma, D. D. in *Nanomaterials chemistry* 139-170 (Wiley-VCH Verlag GmbH & Co. KGaA, 2007).
- 7 Wu, J., Gao, W., Yang, H. & Zuo, J. M. Dissolution kinetics of oxidative etching of cubic and icosahedral platinum nanoparticles revealed by in situ liquid transmission electron microscopy. *ACS Nano* **11**, 1696-1703 (2017).
- 8 Hauwiller, M. R. *et al.* Gold nanocrystal etching as a means of probing the dynamic chemical environment in graphene liquid cell electron microscopy. *J. Am. Chem. Soc.* **141**, 4428-4437 (2019).
- 9 Ye, X. *et al.* Single-particle mapping of nonequilibrium nanocrystal transformations. *Science* **354**, 874-877 (2016).
- 10 Hauwiller, M. R. *et al.* Unraveling kinetically-driven mechanisms of gold nanocrystal shape transformations using graphene liquid cell electron microscopy. *Nano Lett.* **18**, 5731-5737 (2018).

REVIEWERS' COMMENTS:

Reviewer #1 (Remarks to the Author):

In the revised manuscript, the authors addressed the reviewers' comments satisfactorily. Therefore, the manuscript can be accepted for publication in Nat.Commun.

Reviewer #2 (Remarks to the Author):

The authors have satisfied my original critique.

Reviewer #3 (Remarks to the Author):

In this revision, the authors have adequately addressed my questions, comments, and critical feedback, and I believe the work is acceptable for publication. I will observe that the new STEM-EDX maps of the Cu@Au nanocages (Figure S11) do not seem too informative relative to the STEM-EELS already included in Figure 6a and apologize for the experimental difficulty and time spent in acquiring this additional data.